# Digital process control of multi-step assays on centrifugal platforms using high-low-high rotational-pulse triggered valving

**Philip L. Early**[1,2☯], **Niamh A. Kilcawley**[1,2☯], **Niamh A. McArdle**[1,2], **Marine Renou**[1,2,3], **Sinéad M. Kearney**[1,2], **Rohit Mishra**[1,2], **Nikolay Dimov**[1,2], **Macdara T. Glynn**[1,2], **Jens Ducrée**[1,2], **David J. Kinahan**[1,4]*

**1** School of Physical Sciences, Dublin City University, Glasnevin, Dublin, Ireland, **2** National Centre for Sensor Research, Dublin City University, Glasnevin, Dublin, Ireland, **3** Telecom Physique Strasbourg, Université de Strasbourg, Strasbourg, France, **4** School of Mechanical and Manufacturing Engineering, Dublin City University, Glasnevin, Dublin, Ireland

☯ These authors contributed equally to this work.

* david.kinahan@dcu.ie

**Data Availability Statement:** All relevant data are within the paper and its Supporting information files.

## Abstract

Due to their capability for comprehensive sample-to-answer automation, the interest in centrifugal microfluidic systems has greatly increased in industry and academia over the last quarter century. The main applications of these "Lab-on-a-Disc" (LoaD) platforms are in decentralised bioanalytical point-of-use / point-of-care testing. Due to the unidirectional and omnipresent nature of the centrifugal force, advanced flow control is key to coordinate multi-step / multi-reagent assay formats on the LoaD. Formerly, flow control was often achieved by capillary burst valves which require gradual increments of the spin speed of the system-innate spindle motor. Recent advanced introduced a flow control scheme called 'rotational pulse actuated valves'. In these valves the sequence of valve actuation is determined by the architecture of the disc while actuation is triggered by freely programmable upward spike (i.e. Low-High-Low (LHL)) in the rotational frequency. This paradigm shift from conventional 'analogue' burst valves to 'digital' pulsing significantly increases the number of sequential while also improving the overall robustness of flow control. In this work, we expand on these LHL valves by introducing High-Low-High (HLH) pulse-actuated (PA) valving which are actuated by 'downward' spike in the disc spin-rate. These HLH valves are particularly useful for high spin-rate operations such as centrifugation of blood. We introduce two different HLH architectures and then combine the most promising with LHL valves to implement the time-dependent liquid handling protocol underlying a common liver function test panel.

## 1. Introduction

Increasingly over the past decade, centrifugal microfluidic [1–3] systems have been developed for a variety of application fields such as biomedical diagnostics [4, 5], bioprocess monitoring [6] and environmental surveillance [7–10]. The disc-shaped cartridges, which have typical dimensions akin to commonly available optical storage media such as CDs or DVDs, are

**Funding:** JD Science Foundation Ireland under Grant No 10/CE/B1821, Enterprise Ireland under Grant No CF/2011/1311 European Union under Grant No. FP7-KBBE-2013-7-613908-DECATHLON The funders had no role in study design, data collection and analysis, decision to publish, or preparation of the manuscript.

**Competing interests:** The authors have declared that no competing interests exist.

rotated by a compact instrument featuring a simple spindle motor without the need for cumbersome and error-prone pressurised fittings or external pumps. Their inherent capability to centrifuge samples is extremely useful for implementing Laboratory Unit Operations (LUOs) [1] for blood processing [11] and particle / cell handling [12–14]. After loading, these Lab-on-a-Disc (LoaD) devices can be designed to process and analyse the sample in a fully automated fashion, thus making them particularly useful for decentralised testing, e.g. in point-of-care scenarios. However, as all liquids on-disc are simultaneously subjected to the centrifugal field, flow control elements, such as valves, have become fundamental orchestrating a network of LUOs such as on-disc mixing, metering, and reagent release.

Valving techniques on the centrifugal platform can be categorised into externally actuated, or active [15], and rotationally controlled schemes. In addition, a newer class of valves, which we term 'event-triggered', is gaining prominence [16, 17]. Externally actuated valves can be categorised as those where a peripheral instrument (other than the platform-innate spindle motor) transfers energy to the disc. The range of interactions that have been implemented include connection to external pressure sources [18], physical manipulation [19, 20], thermal energy to induce phase-changes [21–24] or even using secondary rotation of the chip to change the relative direction of the centrifugal field [25–28]. While these approaches clearly expand the capabilities of the centrifugal platform, they come at the expense of more complex instrumentation.

The more ubiquitous, rotationally actuated valves open upon variation of the spin rate. This type of valve is based on unbalancing the hydrostatic equilibrium between rotationally induced hydrostatic pressure and other (counter)acting forces on liquid elements such as pneumatic pressure or capillary action. The high-pass version of these valves yield upon exceeding a certain spin rate; they include capillary burst valves [29, 30], centrifugo-pneumatic dissolvable-film (DF) valves [31, 32], elastomeric membranes [33] and dead-end pneumatic chambers [34]. On the other hand, low-pass valves open upon a reduction of the rotational frequency. These flow control elements include conventional siphons primed by capillary action [35, 36] and pneumatically enhanced centrifugo-pneumatic siphon valves (CPSVs) [11, 37–40].

The primary drawback of rotationally actuated valves remains the poor definition of burst frequencies. Variations from the design burst frequency are due to the valves' susceptibility to effects such as contact angle, edge sharpness and geometrical fidelity. Therefore, to implement a typical liquid handling protocol, the burst frequencies of successive valves must be widely separated by large increments to avoid actuation out of sequence. This division of the spin envelope into burst 'bands', combined with a maximum spin rate constrained by factors such as safety, sealing of the disc, and finite power of the spindle motor, limits the number of consecutive valving operations which can be automated. To some extent, this issue has been circumvented by combining low-pass and high-pass valves in series [35] or through optimisation of disc architecture [41].

Recently, we introduced a new type of valve called a rotational pulse actuated dissolvable film valve [42]. In this work, we created dissolvable film valves actuated by a rapid low-high-low (LHL) spike in the rotational disc spin-rate. The order in which the valves were actuated was dependent on the disc architecture while timing of the pulse determined the timing of valve opening. However, a drawback of these valves is unit operations which function best at high spin-rates, such as blood centrifugation, are limited by the actuation spin frequency of these LHL valves. Towards removing this limitation, in this work we introduce 'high-low-high' (HLH) pulse actuated valves. These valves remain closed at high spin-frequencies and are opened by reducing the speed of the disc spin-rate and then increasing it again. We introduce two versions of these valves, one based on the standard siphon valve, and one which uses a combination of a pneumatic chamber with dissolvable films. We then demonstrate how these HLH valves can be combined with LHL valves to automate a bioassay.

## 2. Experimental

### 2.1 Disc manufacture

The discs used in this study were manufactured using xurography [43, 44] as previously described in a multilayer architecture from layers of Poly(methyl methacrylate) (PMMA) and layers of Pressure Sensitive Adhesive (PSA) [16]. Microchannels were patterned in PSA using a knife cutter (CraftRobo Pro, Graphtec, USA) while larger reservoir structures are created by voids in the PMMA (1.5 mm thick) using a laser cutter (Epilog Zing, USA). For their integration into the disc during manufacture, the DFs mounted on PSA to improve sealing and to enhance their mechanical stability during assembly [16]. Two grades of DFs are used in this study. One film, KC 35 (SOLUBLON®, Aicello Corporation, Japan) exhibits a dissolution time of ~40 s in the presence of DI water while the second grade, a low-cost "E-film" used for embroidery (Barnyarns, Rippon, UK) opens within ~6 s [16]. Generally, KC35 film was used where time was required for the motor to return to the nominal spin rate; for swift response to system inputs, we chose fast dissolving E-film. During assembly of discs employing capillary primed siphons, a hydrophilic surfactant (Sigma Aldrich TWEEN® 80 P/N 1754) is applied to each siphon crest by a fine-tipped brush.

### 2.2 Experimental test stand

Images were acquired from a "spin stand" [16, 45] where a spindle motor (Faulhaber Minimotor SA, Switzerland) is synchronised with a stroboscopic light source (Drelloscop 3244, Drello, Germany) and a sensitive, short-exposure time camera (Pixelfly, PCO, Germany). A frame sequence where the disc appears stationary is acquired by capturing images at a set point during each rotation to permit high-resolution observation of liquid flow about the disc.

### 2.3 Absorbance measurements

As the capability to make absorbance measurement from LoaD has been widely demonstrated [9, 46, 47], it was not the primary focus of this work. Absorbance measurements from the liver function assay were made using a commercial plate reader (TECAN Infinite® 200 PRO) from transparent, flat bottomed micro-titre plates (Greiner) loaded with 120 µL sample. Calibrant reagent was also processed on-bench at using the reduced volumes similar to those reported previously [46, 48].

## 3. Valve design and operation

The two different valve types presented in this paper can be modelled by considering behaviour of trapped gas under pressurisation, as idealised using Boyle's Law, where this pressurisation is due to the centrifugally induced hydrostatic pressure applied by a liquid column to the trapped gas as previously described [16]. As previously defined for dissolvable film-based event-triggered valves, these valves have are composed of at least two DFs, one a 'load film' (LF) through which liquid is released, and one a 'control film' (CF) which vents the pneumatic chambers (and so lets the liquid enter the valve and wet and dissolve the LF). The HLH siphon valves must can be modelled in two different states, where the CF is intact and where it is dissolved (where the valve is in a readied and non-readied state respectively. The HLH DF based valves are better modelled as two separate structures which interact to release samples and reagents at appropriate times. The behaviour of DF valves across a range of operating conditions has previously been modelled [49–54]. Note that, in general that while the siphon derived valves will be dependent on contact angle to function properly, both valve types will operate well with a range of different reagents. This is not a property of these valves per se, but rather

an advantage of the Lab-on-a-Disc platform whereby adjusting the spin frequency can change pumping force to compensate for moderate variations in liquid properties (e.g. viscocity). However it should be noted that both valves use water dissolvable films and so the reagents must be largely aqueous [48, 55].

### 3.1 Siphon derived HLH valve

In the siphon derived HLH valve, the geometry of the event-triggered valve is modified to include a capillary siphon between the restrained liquid and the LF (Fig 1). With the CF sealed, trapped air prevents capillary priming of the siphon at low spin rates. As the CF is dissolved at high spin rates, the centrifugal force prevents priming of the capillary siphon. By slowing rotation (with the CF dissolved and the valve thus in a 'readied state'), the siphon is primed, and the valve opens. Due to the multi-layer manufacturing method used for these siphon-derived valves, the recessed LF is only wetted when the disc returns to the elevated spin rate. Thus, the siphon primes during the downward pulse in spin rate and is opened on the 'rising edge'.

Fig 1 shows a structure where three such valves are arranged in sequence. The first valve is vented to atmosphere and thus behaves like a conventional siphon. This siphon primes upon reduction of the disc spin rate; however, the air trapped in the gas pocket prevents priming of the two subsequent valves. When the spin rate is increased, the first valve opens on the 'rising edge'. The subsequent pulses trigger the valves in sequence. The siphon architecture was designed through iteration informed on a review of previous literature pertaining to this valve type [35, 36]. Note that a panel-by-panel description of valve actuation is provided in S1 Fig and movie of valve operation is provided in S1 Video.

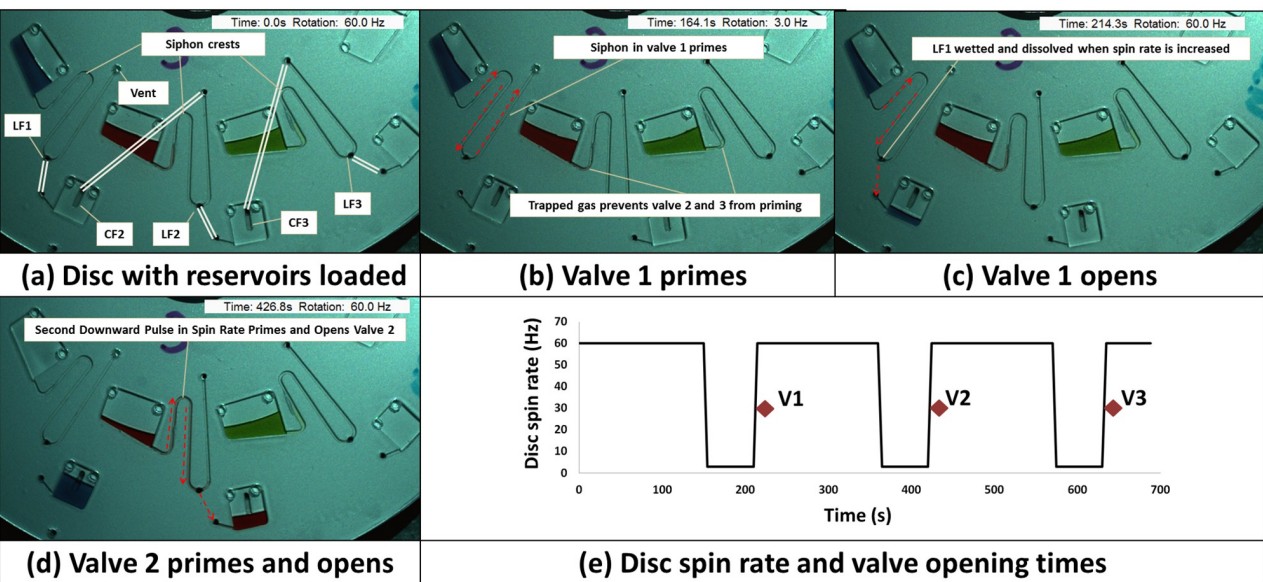

**Fig 1. Cascading of capillary-primed centrifugo-pneumatic siphon valve.** (a) Disc in its initially loaded state. Reservoirs 1, 2, and 3 contain blue, red and green coloured water. Linking channels on the lower microchannel layer are indicated. Valves 2 and 3 are sealed by DFs (E-Film) while the Valve 1 outlet is sealed by an LF but is otherwise open to atmosphere. Note also the siphon crests have been treated with surfactant to aid siphon priming. (b) The disc is decelerated and the first siphon primes (Valve 1). Air trapped in Valve 2 and Valve 3 prevents these siphons from priming (c) Increasing the disc spin rate releases the blue food dye through Valve 1. In every case, the valve actuates on the upward edge of the triggering pulse. This is because, at low spin rates, the recess into which the LF is placed acts as a capillary stop either preventing the LF from being wetted until the spin rate increases or, if the LF is dissolved, acting as a conventional capillary burst valve. (d) Now open to atmosphere, Valve 2 is triggered by the second downward pulse. Valve 3 is also opened in a similar way on the third downward pulse. (e) The spin-profile used to open the three siphon derived valves in series. The opening time of the valves is indicated by the diamond symbols. These valves open within 5 s of the 'rising edge' of the pulse (n = 3, error bars not visible due to scale). See also S1 Fig and S1 Video.

## 3.2 DF derived HLH valves

The second HLH valve (Fig 2) are best described as an event-triggered DF valve where the CF is recessed in a dead-end pneumatic chamber (which is in turn sealed by a second CF referred to as CF*). The system utilises a mechanism similar to a CPSV [11] where increasing the disc spin-rate can lower the liquid level in a chamber and decreasing the spin-rate can increase the liquid level. Again, the valves can best be described in two separate states—where the CF* is intact or where the CF* is dissolved. With CF* intact, when the disc is at high spin-rates the liquid level is below the CF. At low spin-rates, the liquid level will move radially inwards but will not wet the CF due to the air trapped in the dead end chamber. However, with CF* dissolved, this chamber is now vented so at low spin-rates the CF will be wetted and the valve opened.

Fig 2 shows a disc where, for the first valve, CF1 is recessed in a chamber which is evented to atmosphere. On slowing the disc, CF1 is wetted and so liquid is released through LF1. However, both CF2* and CF3* are intact so CF2 and CF3 are not wetted. As the liquid released through LF1 now wets an dissolves CF2*, on the next HLH pulse CF2 can be wetted opening the next chamber. The valve geometries were informed by the theoretical analysis of DF valve geometries and centrifuge-pneumatic principles which have bene previously elucidated [11, 16, 31]. Note that a panel-by-panel description of valve actuation is provided in S2 Fig and movie of valve operation is provided in S2 Video.

# 4. Integrated pulse actuated disc

## 4.1 Liver assay protocol and measurement

Previously Nwankire *et al*. [46] implemented a 6-parameter liver assay protocol; one parameter was determined directly from blood plasma while two parameters, Alkaline Phosphatase

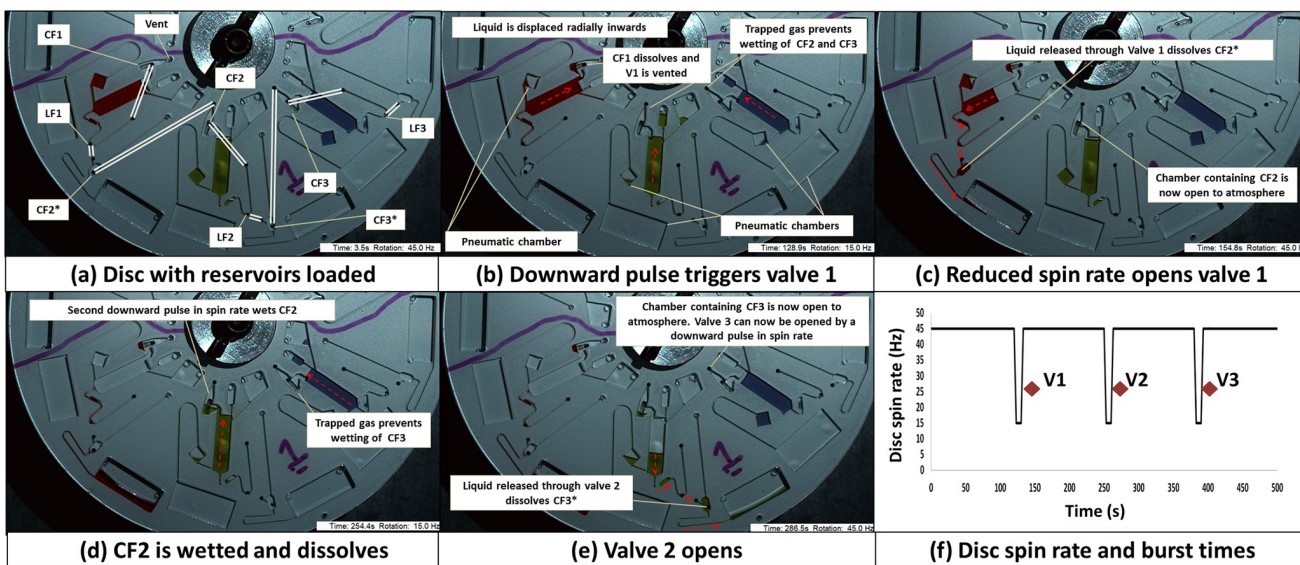

**Fig 2. Low-pass, DF derived pulse-actuated valve.** (a) Disc in its initially loaded state. Reservoirs 1, 2, and 3 contain red, green and blue dyed water. Linking channels on the lower microchannel layer are indicated. In this configuration, the LFs feature slowly dissolving KC-film while E-films are used for CF and CF* tabs. (a) At high spin rates the liquid level is below the CFs for every valve. (b) The disc is decelerated and the liquid elements are displaced upwards due to the expansion of trapped air. CF2 and CF3 are not wetted due to trapped gas. However, CF1 can be wetted and the valve is event-triggered. (c) LF1 is wetted and dissolves. The liquid is released through Valve 1 and then wets and dissolves CF2*; venting the pneumatic chamber which contains CF2. (d) The spin rate is reduced. CF2 can now be wetted; triggering Valve 2. (e) LF2 is wetted and dissolves. The liquid is released through Valve 2 and wets and dissolves CF3*; venting the pneumatic chamber which contains CF3. Thus Valve 3 can be actuated by a subsequent downward pulse in the spin rate. (f) The spin rate profile used to open the three low pass DF valves. The opening time of the valves is indicated by the red diamonds. These valves open within 40 s of the 'rising edge' of the pulse (n = 3, error bars not visible due to scale). See also S2 Fig and S2 Video.

(ALP) and Total Bilirubin (TB) required the addition of a 'stop reagent' prior to an end-point measurement. The remaining three parameters used kinetic or end-point measurements without need to add 'stop reagents'. However, due to the limitations in available burst frequency bands on their disc, Nwankire *et al.* [46] needed to modify one parameter, TB, to a 3-minute incubation prior to the addition of the stop reagent (i.e. identical to the period required for ALP), rather than the 7-minute incubation as specified in the benchtop protocol.

Here, with the innovation focus on the enhanced control enabled by the new pulse actuated liquid handling paradigm, we develop a disc architecture which permits accurate timing of three (parallel) assay protocols. For Direct Bilirubin (DB), where an endpoint measurement must be made after 15 minutes, ALP, where the reaction is stopped after 3 minutes, and TB where the reaction is stopped after (the prescribed) 10 minutes (similar to the that previously described by Kilcawley *et al.* [48]). This liquid handling protocol, as well as the corresponding profile of the spin rate, is outlined in Fig 3. The disc is tested using liver assay calibrant (Randox Laboratories, United Kingdom; Cat. No. CAL2350); the results were compared to benchtop experiments.

## 4.2 Disc architecture

The disc architecture can be split into three sections. The first a pre-processing (blood processing) structure using a DF-derived HLH valve. The metering structures use standard DF burst valves [32] to split the sample into three valves, and mixing with reagents uses LHL pulse-actuated DF valves. Typically, discs manufactured from laminated PSA / PMMA pump reagents reliably at disc spin-rates above 15 Hz and most literature uses speeds of 20–30 Hz [16, 31, 32, 42, 46, 55]. Due to this, we operated the disc at 30 Hz to pump reagents but actuated the valves by reducing the disc spin speed to 15 Hz or increasing it to 60 Hz. The disc architecture and operation is shown in Fig 3 (with detailed panel-by-panel descriptions in S3–S5 Figs and as videos in S3 Video.

**4.2.1 Pre-processing (Blood processing structure) structure.** A centrifugation chamber may be used to extract plasma from whole blood for the liver assay panel. Nwankire et al [46] previously reported this process taking 5 minutes from similar disc architecture with the disc rotating at 33 Hz. In the results reported in this study, only calibrant is used in place of whole blood. This chamber is gated using a Low Pass, DF Derived PA Valve located halfway along its structure. This valve permits centrifugation of sample at high spin rates and then opens with a rotational downward pulse. This is compatible with the LUOs which occur later in the liquid handling protocol.

First, the chamber is loaded in stages with 640 μl of calibrator (S3 Video). The valve stays closed below spin rates of 60 Hz. After processing is complete, the frequency is lowered to 15 Hz. The plasma is displaced upwards, wetting the CF and venting the pneumatic valve. The sample now contacts the LF to open the valve. After this negative pulse, the disc is accelerated to the design operating frequency of 30 Hz.

**4.2.2 Sample metering / aliquoting.** The calibrant released from the centrifugation structure flows down a microchannel and is aliquoted into three chambers. Excess sample flows to the plasma overflow chamber. A DF burst valve (which did not open at the earlier high spin rates as the metering structure did not contain sample) is located at the base of each metering chamber. Metering was characterised using food dye / absorbance measurements, using a method previously described [56], and found to transfer 22.0 ± 0.4 μl (n = 9) into the next chamber (rounded to 22.0 μl for use in this assay).

**4.2.3 Reagent storage, disc operation and results.** The DB reagent is stored on the periphery of the disc in Read Chamber 1 (RC1), the ALP reagent in the Mixing Chamber 1

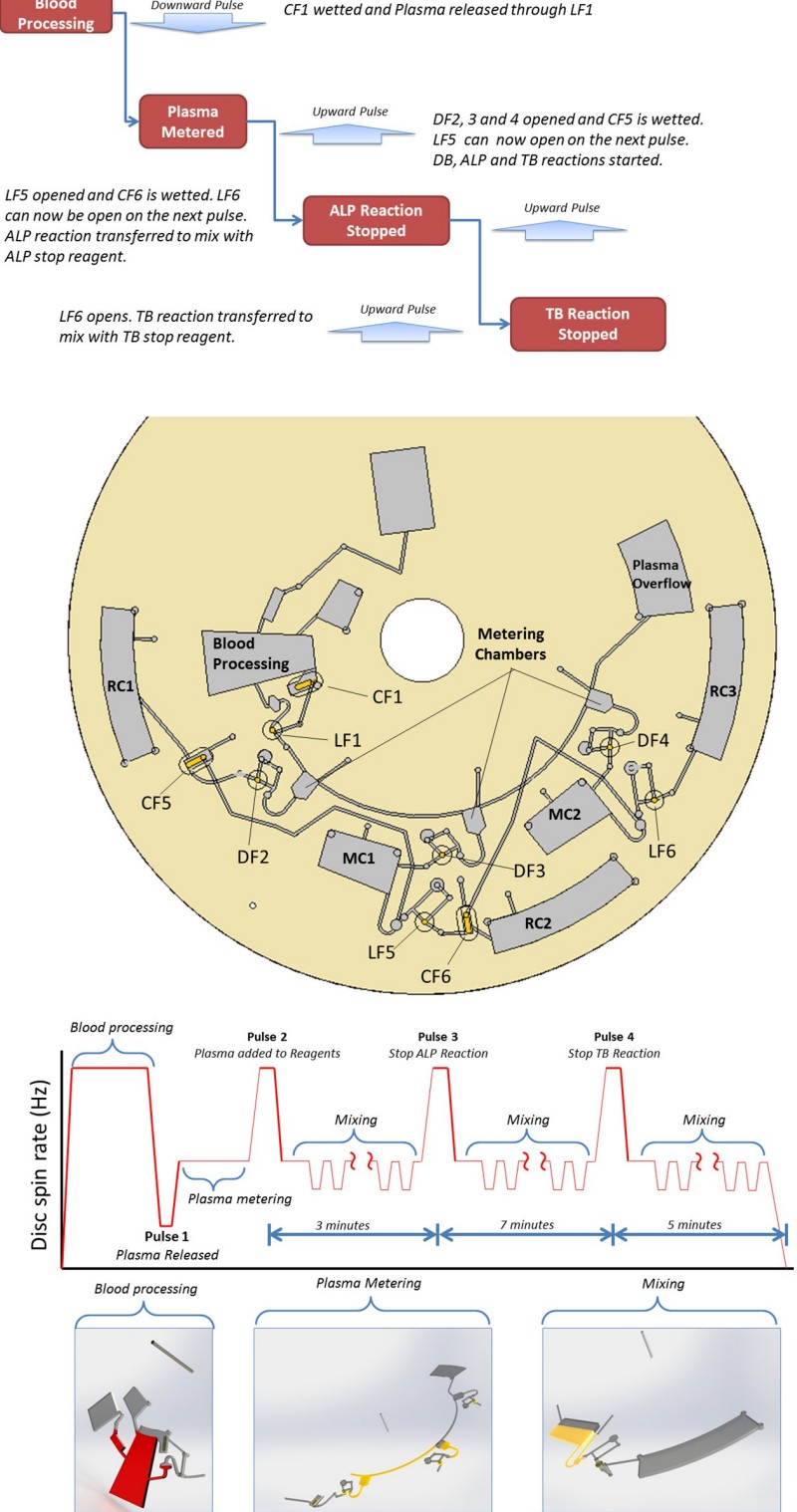

**Fig 3. Liver assay workflow with disc architecture and the spin protocol.** See S3 Video.

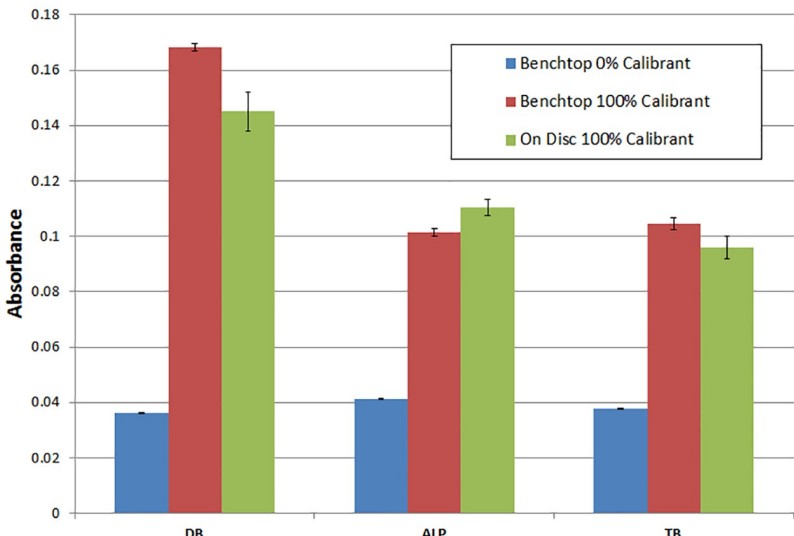

**Fig 4. Comparison of benchtop results (n = 5) versus sample processed on-disc (n = 3).** The 0% calibrant, i.e. negative control, was DI water. The 100% calibrant is 21.2 μmol L$^{-1}$ DB, 127 U L$^{-1}$ ALP and 29.2 μmol L$^{-1}$ TB.

(MC1), the ALP stop reagent in the peripheral Read Chamber 2 (RC2), the TB reagent in Mixing Chamber 2 (MC2) and the TB stop reagent in the peripheral Read Chamber 3 (RC3), respectively.

After discretising the sample into 22-μl volumes, the disc is accelerated to 60 Hz for 12 seconds. This step opens the DFs at the base of the metering chambers and forwards the samples through the disc. This sample transfer also wets LF5, which puts CF5 (which is located at the base of MC1) into a condition where it will open a subsequent pulse. The reactions are then agitated using rapid 'shake mode' mixing. After 3 minutes, another 60-Hz pulse over a 12-second interval opens the valve on MC1 and the ALP reaction is pumped to an outer chamber where it mixes with its stop reagent. The transfer of this reagent wets CF6 with puts LF6 in a 'readied state'. Again, 'shake mode mixing' is applied and, after a further 7 minutes, the spin rate is pulsed to 60 Hz to centrifugally transfer the TB reaction product through LF6. The samples are then agitated using 'shake mode' for a further 5 minutes before (off-disc) measurement.

The samples processed on disc showed good agreement with the samples processed using benchtop protocols (Fig 4); variations are likely attributed to volume inaccuracies and may thus be resolved by higher-end manufacturing techniques such as milling or injection moulding. Importantly, the timing and order of valving operation was highly controlled through the programmable spin-profile. Valve actuation was observed to vary only by ± 5s between individual tests.

## 5. Conclusion and outlook

In this paper we have expanded the toolkit of available DF valves by introducing 'HLH' low-pass pulse actuated valves to complement our existing LHL pulse actuated valves [42]. Due to its actuation by intermittent, 'digital' pulses rather than 'analogue' increments, the maximum number of (sequential) LUOs that can be coordinated on the disc is now only limited by its surface area. Another decisive benefit of digital pulse actuation is its significantly enhanced

forgiveness to the manufacturing tolerances which are inherent to common, economically viable polymer mass manufacturing schemes for microfluidic systems.

Results from this system demonstrated good agreement between the 100% calibrant measured 'on-disc' and 'on-bench'. As a next step, the performance of this system would be improved through further miniaturisation to integrate three separate processing structures to test patient samples. These would process a minimum of two calibrant concentrations (as recommended for clinical / hospital laboratory) to develop a calibration curve and test whole blood in the third chamber. It should be noted that Nwankire et al. [46] have demonstrated this assay from blood using a Lab-on-a-Disc with a portable reader instrument and reported close correlation between on-disc results and benchmark data (from a hospital clinical lab). Of the two valves introduced here, the siphon derived valves require a hydrophilic surface (or hydrophilic surface coating) to function. They are therefore more dependent on the disc material, disc manufacturing fidelity, and the reagent / material interfacial tension (contact angle) than the DF-derived low-pass valves. However, while in their basic form the siphon-derived valves are dependent on contact angle for reliable performance, this architecture can be combined with the centrifugo-pneumatic siphon valve [11] architecture to allow the valves function with mildly hydrophobic reagents. Furthermore, the siphon derived valves are conceptually simpler than the DF-derived architecture, use up less disc space, and have a lower dead-volume than the DF-derived low-pass valves. They also require only two DF valves to function rather than three which will reduce assembly time.

Conversely, the DF-derived low-pass valves operate largely independently of reagent properties and contact angles. They are particularly suitable for blood processing as plasma will dissolve the CF films. However, in contrast to the siphon derived vales they do typically use up more space on the disc due to the use of a pneumatic chamber. In addition, while siphon derived valves can also be used in conjunction with metering / overflow chambers, the DF-derived valves, with their dependence on the liquid displacing inwards to wet the CF film, cannot operate when connected to an overflow metering structure. This drawback means that for applications where high accuracy is required, or user error is likely, they would need to be coupled with a downstream metering structure which will take up more space on the disc.

Importantly, as both valve types employ the same principles and manufacturing techniques as other DF valves (i.e. requiring pick and place of DF tabs) they can both be strategically combined with LHL valves to comprehensively automate bioassays in a sample-to-answer fashion.

## Supporting information

**S1 Fig. 2D Schematic of the operation of the siphon derived low-pass pulse actuated valve.** Here, with the valve pneumatically sealed by the CF valve, the siphon will not prime at low or high centrifugation rates. When the CF is wetted and dissolved, the valve is vented to atmosphere and so the siphon will prime at low disc spin rates. An advantage is that, due to the manufacturing process, the DF is recessed from contact with the liquid. Therefore, the LF is wetted on the 'rising edge' of the downward pulse in spin rate. This characteristic increases the temporal accuracy of the valves.
(TIF)

**S2 Fig. 2D Schematic of the operation of the DF derived low-pass pulse actuated valve.** Here, the core of the valve is a conventional event-triggered valve. This is composed of a Load Film (LF) and a Control Film (CF). The valves functions as the liquid height in the reservoir is controlled by a pneumatic chamber. The CF is recessed in a dead-end pneumatic chamber so that liquid cannot reach it. This pneumatic chamber is sealed by a third dissolvable film, called CF*. At high spin rates, the liquid level is below the CF. At low spin-rates, the liquid can be

displaced inwards but cannot wet the CF due to the trapped gas in the recessed pneumatic chamber. However, with CF* wetted and dissolved, the CF can be wetted and so the valve can be opened.
(TIF)

**S3 Fig. Schematic describing operation of the blood processing structure using for the liver assay panel.** Blood is loaded at a high spin rate (60 Hz). A portion of blood is compressed into a dead end pneumatic chamber. Following separation the spin rate is reduced. This liquid in the pneumatic chamber, now primarily composed of plasm), is pumped back into the main chamber and the liquid height increases. The overflow of plasma wets the CF film. When this dissolves, the valve is vented and the plasma is released form the blood processing structure. This valve is an implementation of the DF-derived low pass pulse actuated valves.
(TIF)

**S4 Fig. Schematic of the metering structure used to separate plasma into three separate aliquots.** Following metering, an upward digital pulse in the spin rate pushes liquid into the dead end pneumatic chambers (DF Valves) where plasma wets the LFs. The chamber is designed so that, on a reduction of the spin rate, the liquid is trapped in contact with the DF. This improves valve reliability when opened via pulses in spin rate. Additionally, the outgoing plasma wets and dissolves the CF which controls the next process valve. Thus, the next valve is placed in a 'Ready State' and can be opened by a pulse in spin rate. This valve is an implementation of DF burst valves described previously in literature.
(TIF)

**S5 Fig. Schematic showing mixing of plasma with reagent.** Mixing of plasma with reagent can be improved by rapid 'shake mode' mixing where the spin rate is robustly increased and decreased. In this schematic, describing the last valve opened in the sequence, when a digital pulse occurs where the valve CF is not dissolved (panel (d)), the valve does not open. However, with this CF dissolved and vented, which occurs as a result of this digital pulse, the valve is in a 'Read State' and will open on the next pulse (panels (f-h). This valve is an implementation of the high pass valves described above.
(TIF)

**S1 Video. Low pass siphons.**
(MP4)

**S2 Video. Low pass dissolvable films.**
(MP4)

**S3 Video. Disc processing calibrant.**
(MP4)

## Acknowledgments

The authors would like to acknowledge and thank Tríona M. O'Connell and Charles E. Nwankire for kind advice regarding the liver assay chemistry; Éanna Bailey, Olivier P. Faneuil and David Kernan for manufacturing some discs used in this study; and Godefroi Saint-Martin for generating selected 3D schematics.

## Author Contributions

**Conceptualization:** David J. Kinahan.

**Formal analysis:** Philip L. Early, Niamh A. Kilcawley, Niamh A. McArdle, Marine Renou, Sinéad M. Kearney, Rohit Mishra, Nikolay Dimov, Macdara T. Glynn, Jens Ducrée, David J. Kinahan.

**Funding acquisition:** Jens Ducrée.

**Investigation:** Philip L. Early, Niamh A. Kilcawley, Niamh A. McArdle, Marine Renou, Sinéad M. Kearney, Rohit Mishra, Nikolay Dimov, Macdara T. Glynn, Jens Ducrée, David J. Kinahan.

**Methodology:** Philip L. Early, Niamh A. Kilcawley, Niamh A. McArdle, Marine Renou, Sinéad M. Kearney, Rohit Mishra, Nikolay Dimov, Macdara T. Glynn, David J. Kinahan.

**Project administration:** Jens Ducrée, David J. Kinahan.

**Resources:** Jens Ducrée.

**Supervision:** Jens Ducrée, David J. Kinahan.

**Writing – original draft:** Jens Ducrée, David J. Kinahan.

**Writing – review & editing:** Philip L. Early, Niamh A. Kilcawley, Niamh A. McArdle, Marine Renou, Sinéad M. Kearney, Rohit Mishra, Nikolay Dimov, Macdara T. Glynn, Jens Ducrée, David J. Kinahan.

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
