## [Decision Letter · Decision Letter 0]

29 May 2023

PONE-D-23-09334Digital Process Control of Multi-Step Assays on Centrifugal Platforms using High-Low-High Rotational-Pulse Triggered ValvingPLOS ONE

Dear Dr. Kinahan,

Thank you for submitting your manuscript to PLOS ONE. After careful consideration, we feel that it has merit but does not fully meet PLOS ONE’s publication criteria as it currently stands. Therefore, we invite you to submit a revised version of the manuscript that addresses the points raised during the review process.

We look forward to receiving your revised manuscript.

Kind regards,

Santiago Casado Rojo, Ph.D.

Academic Editor

PLOS ONE

Journal Requirements:

"This work was supported by the Science Foundation Ireland under Grant No 10/CE/B1821, Enterprise Ireland under Grant No CF/2011/1311 and the European Union under Grant No. FP7-KBBE-2013-7-613908-DECATHLON.."

"JD

Science Foundation Ireland under Grant No 10/CE/B1821, 

Enterprise Ireland under Grant No CF/2011/1311

European Union under Grant No. FP7-KBBE-2013-7-613908-DECATHLON

The funders had no role in study design, data collection and analysis, decision to publish, or preparation of the manuscript"

6. Please upload a copy of Figure 6, to which you refer in your text on page 13 (in PDF format). If the figure is no longer to be included as part of the submission please remove all reference to it within the text.

**Additional Editor Comments:**

As an invited Academic Editor of PLOS ONE, I appreciate being assigned to this interesting manuscript. Please, consider each of the comments submitted by the Reviewers, taking special care of these two expanded here:Variations of media conditions (i.e., temperature, aging, irradiation, etc.), might alter the performance of the system? Either through the modification of the solving kinetics of the fluid, or through its interaction with the PMMA/PSA structure, its functionality may change. Indeed, although it is clear that extreme conditions may induce a complete degradability of the structure itself, could a mild variation also imperceptibly modify its surface, producing a bias? Perhaps a physicochemical analysis of the testing fluid, of the surface of the polymeric structure, and of the interaction between them might help.Why not testing the device with real blood or blood plasma examples? Furthermore, I would rather propose something more: could it be feasible to show a real plasma case where this device may succeed, in comparison to existing ones failing there?

Reviewer's Responses to Questions

**Comments to the Author**

1. Is the manuscript technically sound, and do the data support the conclusions?

Reviewer #1: Yes

Reviewer #2: Partly

2. Has the statistical analysis been performed appropriately and rigorously? 

Reviewer #1: Yes

Reviewer #2: No

3. Have the authors made all data underlying the findings in their manuscript fully available?

Reviewer #1: Yes

Reviewer #2: No

4. Is the manuscript presented in an intelligible fashion and written in standard English?

Reviewer #1: Yes

Reviewer #2: Yes

5. Review Comments to the Author

Reviewer #1: This manuscript mainly demonstrated two different HLH valves designs (Siphon Derived HLH Valve and DF Derived HLH Valves). Controlling by digital pulsing, these valves could achieve a more robust flow control with higher sequential steps. Furthermore, authors integrate the HLH valve with previously developed LHL valves and standard DF burst valves and deploy it to measure DB, TB and ALP based on a liver assay protocol. Here are some technical comments for this manuscript:

1. Authors introduce two different HLH valves: Siphon Derived and DF Derived HLH Valves. It seems that siphon derived valve is highly dependent on the hydrophilicity of the materials, which could has a limited use compared with DF derived one. In the liver assay demonstration, authors didn't use siphon derived HLH valves either. Then it is confusing to show the siphon derived HLH valve in this manuscript unless authors could use experiments to justify its clear advantages or unique properties.

2. In the main figure, authors only show the operation process of HLH valves. However, illustration of he detailed mechanism is missing. Please include more zoom-in schematics to describe what is happening at the place of interest.

3. The photos quality is poor in terms of color contrast and brightness. It is difficult for readers to see the colors and details.

Reviewer #2: In this manuscript, Early et al. introduced a fabrication technique to use dissolvable membrane to fabricate pulse-actuated valves for liquid processing and handling. Overall, the manuscript is well organized, but there are still some issues to address before it can be accepted.

1. In the manuscript the author used calibrants for the assay demonstration, and it is unclear how close it is related to biofluids such as blood. Therefore, how would the device perform if blood was processed in the device?

2. It is unclear about how the properties of the biofluid would influence the processing results. For example, how would the viscosity affect the results. The author should include a detailed investigation targeting different kinds of biofluids.

3. The device optimization is also not clear in the manuscript. How did the author choose the different frequencies for the operation?

4. The efficacy of the device also lacks characterizations such as the filtration rate for the sample.

5. Some supplementary materials were not included in the submission.

6. PLOS authors have the option to publish the peer review history of their article (what does this mean?). If published, this will include your full peer review and any attached files.

Reviewer #1: No

Reviewer #2: No

---

## [Author Response · Author response to Decision Letter 0]

4 Aug 2023

Response to reviewers is uploaded as a separate file.

---

## [Decision Letter · Decision Letter 1]

24 Aug 2023

Digital Process Control of Multi-Step Assays on Centrifugal Platforms using High-Low-High Rotational-Pulse Triggered Valving

PONE-D-23-09334R1

Dear Dr. Kinahan,

We’re pleased to inform you that your manuscript has been judged scientifically suitable for publication and will be formally accepted for publication once it meets all outstanding technical requirements.

Kind regards,

Santiago Casado Rojo, Ph.D.

Academic Editor

PLOS ONE

Reviewers' comments:

Reviewer's Responses to Questions

**Comments to the Author**

1. If the authors have adequately addressed your comments raised in a previous round of review and you feel that this manuscript is now acceptable for publication, you may indicate that here to bypass the “Comments to the Author” section, enter your conflict of interest statement in the “Confidential to Editor” section, and submit your "Accept" recommendation.

Reviewer #1: All comments have been addressed

Reviewer #2: All comments have been addressed

2. Is the manuscript technically sound, and do the data support the conclusions?

Reviewer #1: Yes

Reviewer #2: Yes

3. Has the statistical analysis been performed appropriately and rigorously? 

Reviewer #1: Yes

Reviewer #2: Yes

4. Have the authors made all data underlying the findings in their manuscript fully available?

Reviewer #1: Yes

Reviewer #2: Yes

5. Is the manuscript presented in an intelligible fashion and written in standard English?

Reviewer #1: Yes

Reviewer #2: Yes

6. Review Comments to the Author

Reviewer #1: I appreciate authors' clarification about the advantages of the HLH siphon-derived valve and the new figures layout. The manuscript becomes more transformative to the general scientific communities. Therefore, I recommend acceptance of the manuscript.

Reviewer #2: (No Response)

7. PLOS authors have the option to publish the peer review history of their article (what does this mean?). If published, this will include your full peer review and any attached files.

Reviewer #1: No

Reviewer #2: No

---

## [Editor Report · Acceptance letter]

29 Aug 2023

PONE-D-23-09334R1 

Digital Process Control of Multi-Step Assays on Centrifugal Platforms using High-Low-High Rotational-Pulse Triggered Valving 

Dear Dr. Kinahan:

I'm pleased to inform you that your manuscript has been deemed suitable for publication in PLOS ONE. Congratulations! Your manuscript is now with our production department. 

Kind regards, 

on behalf of

Dr. Santiago Casado Rojo 

Academic Editor

PLOS ONE